# Transcriptomic signatures of WNT-driven pathways and granulosa cell-oocyte interactions during primordial follicle activation

Hinako M. Takase[1☯]*, Tappei Mishina[1,2☯], Tetsutaro Hayashi[3,4], Mika Yoshimura[3], Mariko Kuse[3], Itoshi Nikaido[3,4], Tomoya S. Kitajima[1]

1 Laboratory for Chromosome Segregation, RIKEN Center for Biosystems Dynamics Research (BDR), Kobe, Japan, 2 Faculty of Agriculture, Kyushu University, Fukuoka, Japan, 3 Laboratory for Bioinformatics Research, RIKEN Center for Biosystems Dynamics Research (BDR), Kobe, Japan, 4 Department of Functional Genome Informatics, Division of Biological Data Science, Medical Research Institute, Tokyo Medical and Dental University (TMDU), Bunkyo, Japan

☯ These authors contributed equally to this work.
* hinako.takase@riken.jp

**Data Availability Statement:** All RNA sequencing data have been deposited in the GEO database with accession number GSE255212. All other relevant

## Abstract

Primordial follicle activation (PFA) is a pivotal event in female reproductive biology, coordinating the transition from quiescent to growing follicles. This study employed comprehensive single-cell RNA sequencing to gain insights into the detailed regulatory mechanisms governing the synchronized dormancy and activation between granulosa cells (GCs) and oocytes with the progression of the PFA process. *Wntless* (*Wls*) conditional knockout (cKO) mice served as a unique model, suppressing the transition from pre-GCs to GCs, and disrupting somatic cell-derived WNT signaling in the ovary. Our data revealed immediate transcriptomic changes in GCs post-PFA in *Wls* cKO mice, leading to a divergent trajectory, while oocytes exhibited modest transcriptomic alterations. Subpopulation analysis identified the molecular pathways affected by WNT signaling on GC maturation, along with specific gene signatures linked to dormant and activated oocytes. Despite minimal evidence of continuous up-regulation of dormancy-related genes in oocytes, the loss of WNT signaling in (pre-)GCs impacted gene expression in oocytes even before PFA, subsequently influencing them globally. The infertility observed in *Wls* cKO mice was attributed to compromised GC-oocyte molecular crosstalk and the microenvironment for oocytes. Our study highlights the pivotal role of the WNT-signaling pathway and its molecular signature, emphasizing the importance of intercellular crosstalk between (pre-)GCs and oocytes in orchestrating folliculogenesis.

## Introduction

Folliculogenesis is a strictly controlled and dynamic process that is essential for the success of female reproduction. The formation of primordial follicles occurs perinatally in mice [1], with

data are within the paper and Supporting Information files.

**Funding:** This work was supported by The Japan Society for the Promotion of Science (JSPS) KAKENHI Grant Numbers 21K06193 and 22KJ3149 to H.T.; and Japan Science and Technology (JST) Agency CREST Grant Number JPMJCR21N6 to I.N. The funders had no role in study design, data collection and analysis, decision to publish, or preparation of the manuscript.

**Competing interests:** The authors have declared that no competing interests exist.

characteristic flattened pre-granulosa cells (pre-GCs) encompassing each oocyte individually [2]. Primordial follicles are particularly important in determining ovarian reserves, serving as a pool of dormant oocytes. A subset of primordial follicles undergoes activation, initiating the irreversible process of folliculogenesis. During primordial follicle activation (PFA) to become primary follicles, pre-GCs develop into cuboidal and proliferative granulosa cells (GCs) and simultaneously oocytes start to grow [3, 4]. Subsequent stages of folliculogenesis include secondary follicles, antral follicles and preovulatory follicles, with only a limited number of oocytes reaching ovulation. This continuous process of folliculogenesis is repeated throughout the reproductive period until menopause. Since the failure of PFA can lead to an inadequate or a transiently excessive supply of oocytes, the precise regulation of PFA is imperative for successful reproduction [5, 6]. Anomalies in PFA are believed to contribute to premature ovarian insufficiency in humans [7]. Nonetheless, how primordial follicle quiescence and activation are regulated is incompletely understood.

PFA is an orchestrated process of somatic cells and oocytes, with bidirectional communication between them being crucial. Therefore, it is important to understand the intricate control of the dynamic states of both GCs and oocytes during PFA. To date, several key molecular components that regulate PFA have been identified. The maintenance of oocyte dormancy relies on the nuclear localization of the transcription factor forkhead box O3 (FOXO3), a reliable indicator of PFA [6, 8]. In oocytes, FOXO3 is phosphorylated and translocated to the cytoplasm via the PI3K-AKT signaling pathway followed by KIT signal transduction, in response to GC-derived ligands which leads to PFA [9]. It is also known that oocyte-derived GDF9 promotes GC cell proliferation [10] and the mTORC1 pathway in pre-GCs contributes to PFA [11]. In addition, environmental factors such as hypoxia, extracellular matrix (ECM)-mediated pressure and nutritional factors have been identified as crucial components in the maintenance of primordial follicles [12, 13]. The oocyte surrounding microenvironments provided by pre-GCs are dynamically changed during PFA associated with GC maturation. Despite research efforts on the molecular components that govern PFA, the complexities of GC-oocyte communication and their dynamic states has remained elusive.

We and other research groups have reported the significant role of WNT signaling in postnatal folliculogenesis in mice. Interestingly, according to a study by De Cian *et al.*, the oocyte-GC interaction is also key here, as oocyte-derived RSPO2, a positive regulator of WNT signaling, acts on neighboring GCs [14]. In contrast, our research revealed an autocrine function of WNT signaling in pre-GCs during PFA to permit transformation to GCs. In *Wls* cKO mice, in which somatic cell-derived WNT secretion is suppressed, pre-GCs fail to acquire GC properties upon PFA stimulation, remaining flat, with less cytoplasm and with low cell proliferative activity [15]. Despite the simultaneous activation expected for GCs and oocytes, they failed to synchronize in *Wls* cKO mice—only the oocytes grow and increased in size. However, these oocytes exhibited some immature characteristics such as possible retained dormancy as indicated by the nuclear localization of FOXO3. Since these results suggest that oocyte awakening requires support from functional GCs, *Wls* cKO mice emerge as a valuable tool for investigating the unidentified role of GCs in this context. In this study, we aimed to advance foundational knowledge about folliculogenesis and infertility by uncovering the single-cell resolution regulatory dynamics during PFA using *Wls* cKO mice.

## Results

### Ovaries of *Wls* cKO mice show abnormalities in GCs after PFA

The process of primordial follicle assembly in mice takes place from before birth to postnatal day (PD) 5, followed by the growth of activated primordial follicles starting from PD6 [16–18].

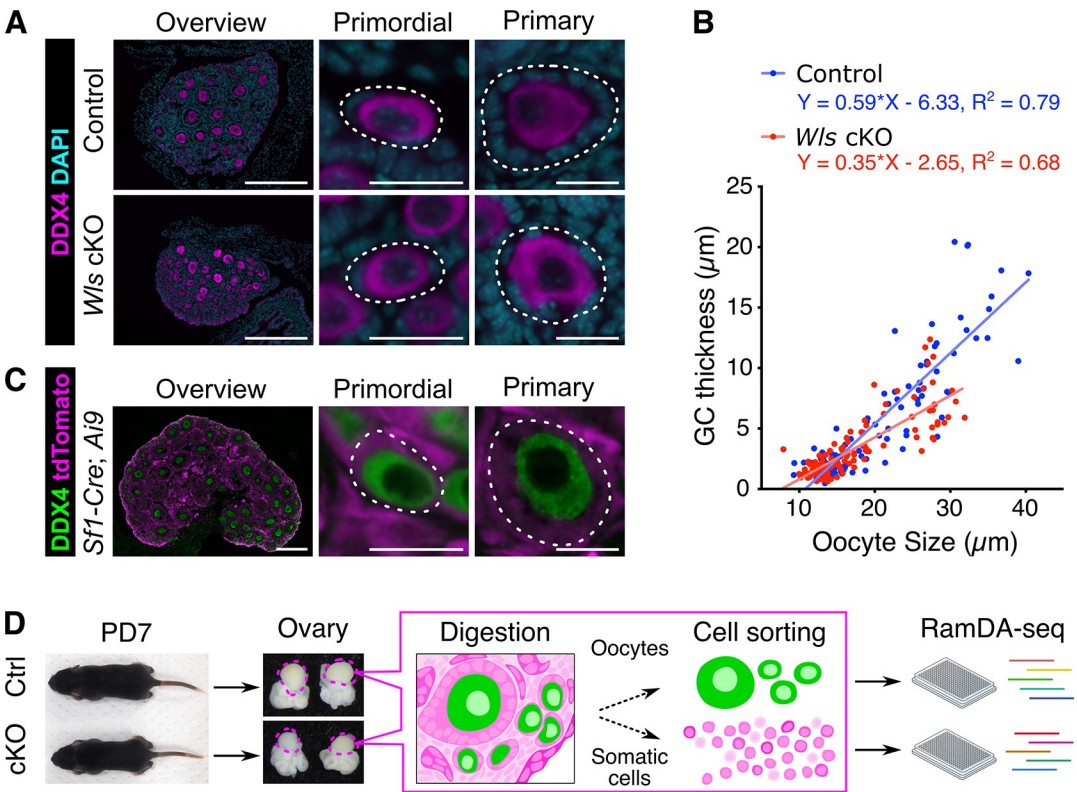

**Fig 1. Suppressed WNT signaling in pre-GCs leads to insufficient differentiation of the GC layer in growing follicles.** (**A**) Immunofluorescence staining of DDX4 (magenta) in the ovaries of *Wls* cKO mice and littermate control mice at PD7. Nuclei were counterstained with DAPI (cyan). White dotted lines indicate follicles. Scale bars, 200 μm (leftmost panels) or 20 μm (other panels). (**B**) Scatter plot for the distribution of oocyte diameter and GC layer thickness for follicles of *Wls* cKO (n = 161 follicles) and control (n = 241 follicles) mice at PD7. Solid lines represent the linear regression fit with equations. (**C**) Immunofluorescence staining of DDX4 (green) and tdTomato fluorescence (magenta) in the ovaries of 2-week-old *Sf1-Cre; Ai9* mice. (**D**) Experimental design for scRNA-seq analysis. Ovaries from *Wls* cKO (cKO, *Sf1-Cre;Wls*flox/del*;Ai9*, n = 8) or littermate control (Ctrl, *Sf1-Cre;Wls*flox/+*;Ai9*, n = 3) mice were collected at PD7 after which single-cell suspensions were obtained. scRNA-seq was performed using RamDA-seq and the data were then analyzed.

Ovarian samples were collected from wild-type and from *Wls* cKO mice at PD7, with an assumption that the cell populations around the transition from primordial to primary follicles were abundantly present. Subsequently, we conducted fluorescent immunostaining for an oocyte marker, DDX4, and confirmed the presence of primordial, primary and small secondary follicles in control ovaries (Fig 1A). In contrast, *Wls* cKO ovaries contained primordial follicles that appeared normal and primary follicles surrounded by abnormal GCs, which was characterized by the absence of cuboidal and organized GC layers. To quantitatively evaluate the phenotype of *Wls* cKO mice at PD7, we examined the thickness of the GC layer as well as the size of oocytes by employing PAS staining of ovarian sections. An analysis of covariance (ANCOVA) detected a significant disparity in GC thickness between control and *Wls* cKO mice. (Fig 1B). These differences primarily reflect changes in the state of follicles after activation, as both regression lines are close for primordial follicles with diameters less than 20 μm. The underdevelopment of the GC phenotype becomes more prominent at a later stage of folliculogenesis [15]. These results indicate that control and *Wls* cKO mice provide contrasting models to investigate the molecular mechanisms of PFA, and that ovaries at PD7 offer an optimal window for examining PFA.

## Single-cell transcriptome profiling of ovarian cells from *Wls* cKO mice and from control mice identify different cell types

To investigate cell type-specific alterations in gene expression during PFA with or without WNT-signaling, we performed scRNA-seq analysis on the ovaries of control mice and *Wls* cKO mice at PD7 using RamDA-seq, a full-length total RNA-sequencing method for single cells [19]. Due to the low oocyte-to-GC ratio and limited cells from our genetically modified mice, a plate-based method was required. We selected the RamDA-seq method for this study due to its high sensitivity and proven effectiveness in analyzing oocytes [20], ensuring sufficient sequencing depth for each cell. For these analyses, to obtain ovarian somatic cells by flow cytometry, we utilized mice expressing red fluorescence (tdTomato) driven by *Sf1(Nr5a1)*-Cre. Our cryosection analysis revealed the ubiquitous expression of tdTomato in ovarian component cells, including GCs, while DDX4-positive oocytes were tdTomato-negative (Fig 1C). tdTomato-positive somatic cells and KIT-positive oocytes were sorted from PD7 *Wls* cKO mice (*Sf1-Cre;Wls*$^{flox/del}$*;Ai9*) and from their littermate controls (*Sf1-Cre;Wls*$^{flox/+}$*;Ai9*) by cell sorting (S1A and S1B Fig). We obtained the high-quality transcriptome of 286 somatic cells and 95 oocytes collected from control mice and from *Wls* cKO mice (Fig 1D). To ensure the accuracy and reliability of our single-cell RNA sequencing data, we performed a rigorous quality control process (S2A–S2C Fig). 721 high-quality cells were retained for downstream analysis, representing 94.5% of the initial cell population. Finally, key quality metrics including the assigned genome rate, mitochondrial gene content, and rRNA percentage were plotted against the PCA plot to confirm the robustness of our QC approach (S3 Fig).

To characterize the ovarian cell type during PFA, the control and *Wls* cKO datasets were analyzed separately. Seurat-based unsupervised clustering and visualization with uniform manifold approximation and projection (UMAP) identified 6 and 7 clusters in the control and *Wls* cKO datasets, respectively, demonstrating substantial differences in those cell populations (Fig 2A and 2B). The top 20 differentially expressed genes (DEGs) for each cluster (Fig 2C, 2D, S1 and S2 Tables) were used to identify corresponding known cell types based on the expression patterns of the included known cell type-specific markers for each cluster on UMAP plots: for oocytes (*Ddx4*), for GCs (*Amh*), for pre-GCs (*Smad3*), for ovarian surface epithelia (OSE) (*Krt19*), for mesenchymal cells (*Col1a1*) and for early theca cells (eTCs) (*Enpep*) (Fig 2E) [21, 22]. Among the *Col1a1*-positive mesenchymal cells, those with a low expression of *Enpep* were considered interstitial cells. These analyses successfully identified five distinct cell clusters in the control group: pre-GCs, GCs, OSE, interstitial cells (Int), eTCs and single oocyte clusters (OOs) (Fig 2A). Consistent with a previous report [15], expression of *Amh* in the GC cluster was lower in *Wls* cKO mice than in the control mice (Fig 2E). Notably, the *Wls* cKO ovaries exhibited an absence of the eTC population, while two additional clusters were present in the GC lineage (altered (alt)-GCs) and in oocytes (altered (alt)-OOs) (Fig 2B). The lack of an eTC population in *Wls* cKO ovaries aligns with the absence of normal developing follicles, as theca cells are typically recruited around the secondary follicle stage. The alt-GC cluster identified in *Wls* cKO samples was characterized by a moderate GC marker expression and DEGs such as *Pdlim4* and *Hsd3b1* (Fig 2D and 2E). The alt-oocyte cluster in *Wls* cKO samples expressed well-known oocyte-specific markers in the DEGs, such as *Zp3*, *Lhx8* and *Gdf9*, closely resembling the overall gene expression patterns of the oocyte cluster (Fig 2D). Nevertheless, it exhibited lower values in various quality assessments, including reduced ratios of reads mapping to the intronic region (percent intron), decreased total reads and fewer detected genes compared to the oocyte cluster (Fig 2E and S1C and S1D Fig) (for details, see the oocyte subpopulation analysis below). Since oocytes in *Wls* cKO mice are genetically normal, their growth is predicted to be hindered by their interaction with abnormal GCs.

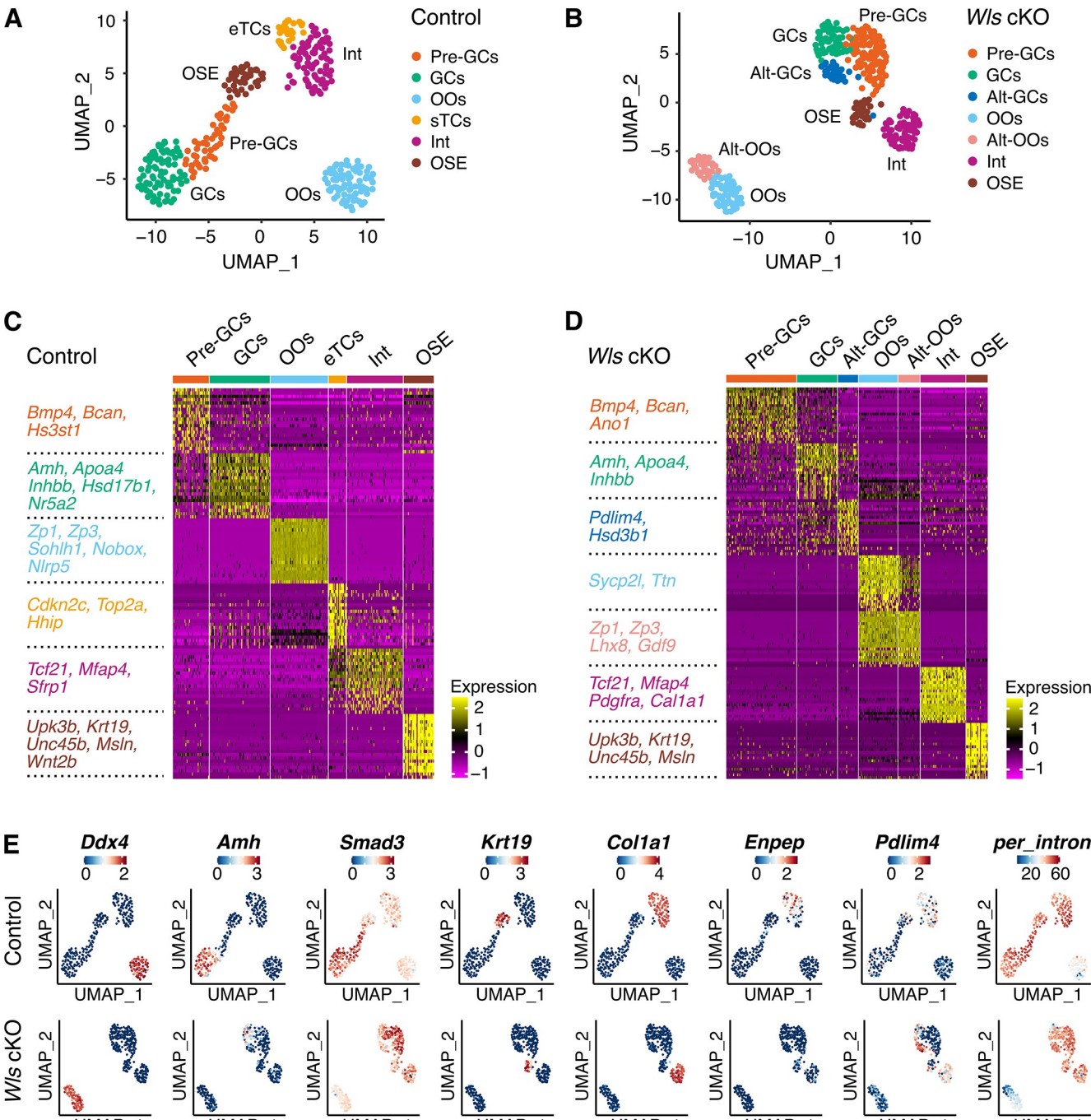

**Fig 2. scRNA-seq of PD7 ovaries reveals a cellular signature derived from *Wls* cKO mouse ovaries.** (**A, B**) Uniform manifold approximation and projection (UMAP) plots of unbiased clustering of ovarian cells from control mice (**A**, *Sf1-Cre;Wls*flox/+*;Ai9*, n = 350 cells) or from *Wls* cKO mice (**B**, *Sf1-Cre;Wls*flox/del*;Ai9*, n = 371 cells), where each dot represents an individual cell that is color-coded based on its identified clusters. (**C, D**) Heatmap of the top 20 DEGs of each cluster for control (**C**) or *Wls* cKO (**D**) samples. The value for each gene is the row-scaled Z-score. Representative DEGs for each cell type are listed on the left. (**E**) UMAP plots showing the expression levels of selected marker genes (*Ddx4*, *Amh*, *Smad3*, *Krt19*, *Col1a1*, *Enpep* and *Pdlim4*) or the ratio of intronic reads (*per_intron*). The color scale represents the gene expression level scaled by the Z-score.

Altogether, we identified six and seven different ovarian cell types for control and *Wls* cKO ovaries, respectively, including altered cell maturation without WNT-signaling in GCs, and characterized the gene expression features of each cell type.

## Distinct subclusters in GC maturation and their molecular signatures

To investigate the GC development process during PFA, we conducted a subpopulation analysis of GCs by integrating data from control (pre-GCs and GCs in Fig 2A) and from *Wls* cKO (pre-GCs, GCs and alt-GCs in Fig 2B) cells. Since our RamDA-seq technique has been known to produce highly reproducible data between batches [19], we integrated two data without any batch corrections to avoid the loss of biological difference between genotypes. In fact, possible artifacts during library preparation were rejected based on UMAP clustering analysis using ERCC RNA spike-in showing a highly mixed genotype-independent distribution (S4A Fig). Furthermore, PCA for whole cells showed that the major PC axes were not associated with genotype (S4B Fig), although detailed exploration found minor effects around PC4 and PC5. These results validated our measure of data integration.

The samples from *Wls* cKO and control ovaries exhibit mostly non-overlapping distributions on the UMAP dimensionality reduction, indicating a genotype-dependent developmental trajectory (Fig 3A). That analysis identified six distinct subgroups with specific DEGs (referred as G1–G6 subclusters), which sheds light on GC lineage development during PFA (Fig 3B–3D and S3 Table). Next, we examined the gene expression patterns across the identified subclusters. The heatmap of the top 20 DEGs for each subcluster (Fig 3C) highlighted distinct molecular profiles, while the dot plots (Fig 3D) illustrated key marker gene expression across subclusters. Below, we provide a detailed description of the defining features of each subcluster, in the order of Control and then *Wls* cKO.

Within the G1 subcluster, *Wls* cKO and control samples were closely positioned and comprised both cell types (Fig 3A and 3B), further indicating the successful integration of the two datasets. Notably, known mature GC markers such as *Inhbb* and *Apoa4* were markedly downregulated in this cluster (Fig 3E). Conversely, *Smad3*, a gene associated with pre-GCs [21], was highly expressed, identifying this subcluster as pre-GCs (Fig 3E). Additionally, a proliferation marker *Top2a* was downregulated in this subcluster, indicating a rather quiescent cell state (Fig 3E). Focusing on the trajectory of the control GCs, the highest expression of the known GC markers *Vcan* and *Amh* in the G3 subcluster led to the assumption they are mature GCs (GCs-Ctrl) (Fig 3D). The G2 subcluster displayed an intermediate gene expression between pre-GCs and GCs-Ctrl, and therefore were presumed to be in the transition state (Transition-Ctrl).

Analogously, the G5 subcluster was designated as GC-cKO due to its higher expression of the known GC markers *Inhbb* and *Apoa4* within the *Wls* cKO GC trajectory (Fig 3E). However, GCs-cKO does not fully acquire the typical transcriptome properties of normal GCs, as indicated by the insufficient expression of *Vcan* and *Amh*, suggesting an aberrant activation (Fig 3D). The G4 subcluster positioned between GCs-cKO (G5) and pre-GCs (G1) was identified as Transition-cKO. Lastly, the G6 subcluster exhibits the lowest expression of the pan-GC markers *Nr5a1* and *Amhr2* (Fig 3E). The underdevelopment of GCs-cKO could lead to the altered GC (Alt-GCs; G6) subcluster characterized by marker genes linked to stress responses, such as *Pdlim4* and *Clu* (Fig 3D). The consistent expression pattern of PDLIM4 was observed by immunostaining of ovarian sections from control and *Wls* cKO mice at PD7 and at 3-week-old (Fig 3G). The results revealed a robust expression of PDLIM4 in GCs of secondary follicles in *Wls* cKO ovaries, while a minimal expression of PDLIM4 was observed in the entire GC lineage of control ovaries. Therefore, the Alt-GC subcluster in *Wls* cKO ovaries likely includes

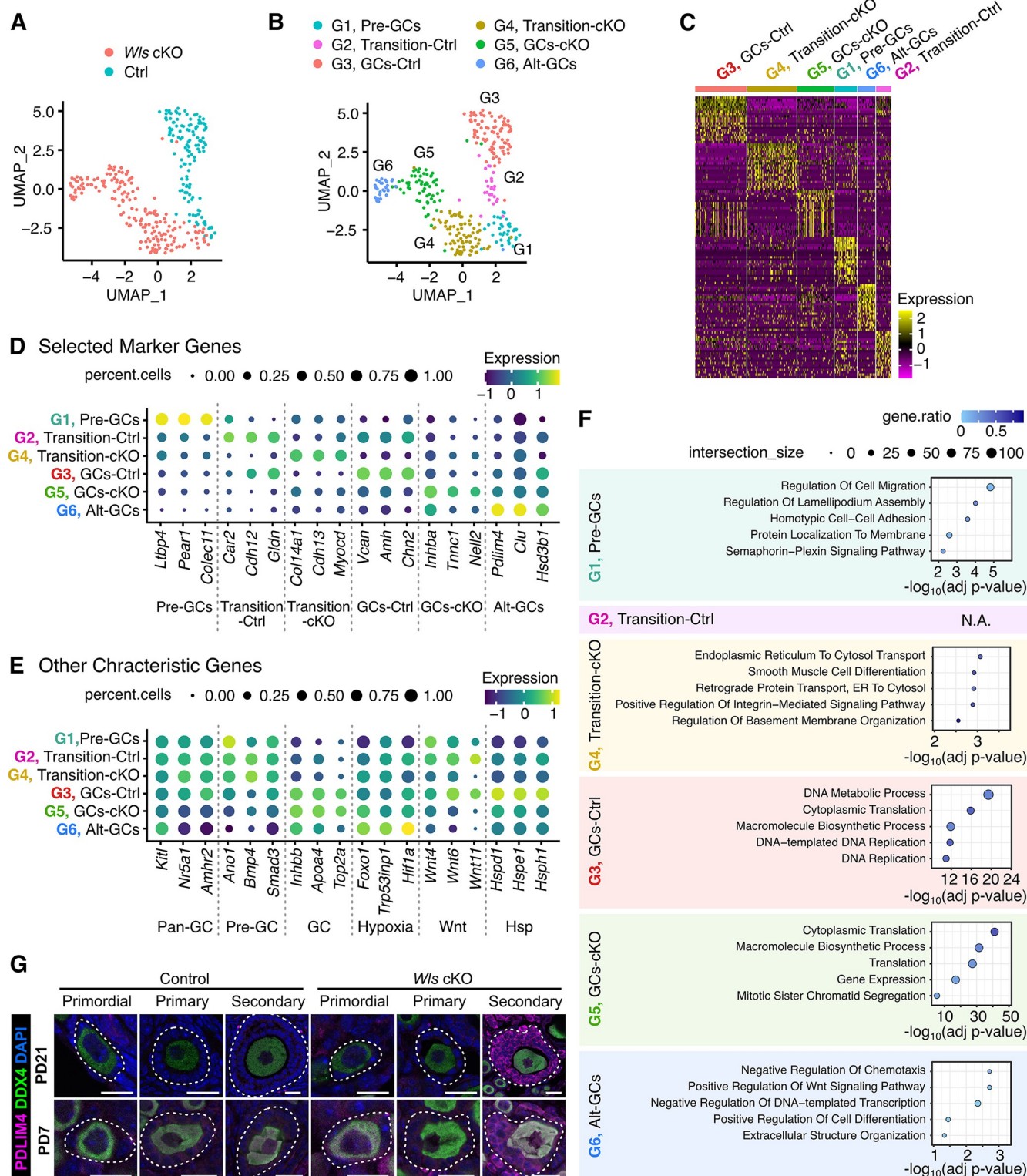

**Fig 3. Integrated subpopulation analysis identifies distinct subgroups in GC lineages.** (**A**) UMAP plot of GCs color-coded by genotype (Ctrl, blue; *Wls* cKO, red). (**B**) UMAP plot showing the different cell subclusters belonging to the GC lineage. (**C**) Heatmap of the top 20 DEGs of each subcluster. (**D, E**) Dot plots showing the expression of selected markers for subclusters or characteristic features. The dot size represents the ratio of cells expressing a specific marker (TPM > 1), while the color indicates the expression level scaled by the Z-score. (**F**) Representative significantly enriched GO terms for DEGs of the GC subcluster. (**G**) Immunofluorescence staining of PDLIM4 (magenta) and DDX4 (green) in the ovaries of PD7 and 3-week-old *Wls* cKO mice and

littermate control mice. Nuclei were counterstained with DAPI (blue). White dotted lines indicate follicles. Scale bars, 10 μm (Primordial) and 20 μm (Others).

GCs from secondary follicles and their subsequent developmental stages. Together, analysis of the subcluster distribution reveals that in *Wls* cKO ovaries, pre-GCs undergo a distinct differentiation trajectory shortly after the PFA.

Note that the annotations for subclusters in this analysis, although some of them share identical names, may not precisely match those in Fig 2 due to the finer classification. By comprehensively examining markers for the broad GC lineage (*Kitl*, *Nr5a1* and *Amhr2*), the pre-GCs as depicted in Fig 2 (*Ano1* and *Bmp4*) and the mature GCs (*Inhbb* and *Apoa4*), we observed an expected but gradual variation between subclusters (Fig 3E). We also observed the anticipated reduced production of WNT ligands (*Wnt4*, *Wnt6* and *Wnt11*), which have been reported to be expressed in GC lineages, in *Wls* cKO subclusters (G3, G4 and G6) [15]. Additionally, we noticed the inclusion of heat shock proteins (*Hspd1*, *Hspe1* and *Hsph1*) specifically in the marker genes of the GC-Ctrl (G3) subgroup, suggesting a potential dependence on WNT signaling for that response.

Next, we proceeded to examine the biological features of each identified subcluster using Gene ontology (GO) analysis of DEGs. That analysis revealed the presence of distinct characteristics of GC cells and illustrated the aberrant maturation pathway in *Wls* cKO ovaries (Fig 3F). For example, the pre-GC (G1) subcluster exhibited GO terms and marker genes linked to cell migration and adhesion, indicating a unique cellular behavior in pre-GCs as a common mechanism in the early stages of GC development. The Transition-Ctrl (G2) subcluster did not yield significant GO terms, primarily due to the limited number of DEGs. Nevertheless, a few marker genes related to calcium regulation and the ECM were identified, such as *Cdh12* (Fig 3D). Although the Transition-cKO (G4) subcluster included GO terms related to the ECM, such as "positive regulation of integrin-mediated signaling pathway", the most prominent term was "endoplasmic reticulum to cytosol transport." This process is thought to be crucial for maintaining cellular homeostasis by eliminating damaged proteins from the ER, suggesting a response to stressful environments. The GC-Ctrl (G3) subcluster exhibited terms linked to cell proliferation and protein synthesis. In contrast, the cell proliferation-related term was not significantly enriched in the GC-cKO (G5) subcluster, which is consistent with our previous findings that *Wls* cKO GCs exhibited reduced proliferation [15]. Nevertheless, the GC-cKO (G5) subcluster showed somewhat similar characteristics to the GC-Ctrl (G3) subcluster, such as the enrichment of terms related to protein translation. The Alt-GC (G6) subcluster corresponded to the Alt-GC cluster in the entire ovarian cell dataset of Figs 2 and 3B). The Alt-GC (G6) subcluster is primarily characterized by the GO terms of suppression of cellular movement and transcription. While not statistically significant in the GO term analysis, it came to our attention that the Alt-GC (G6) subcluster exhibited elevated expression of hypoxia-related genes (*Foxo1*, *Trp53inp1* and *Hif1a*), implying a deteriorating cellular environment (Fig 3E). Collectively, our analysis revealed a pivotal role for WNT signaling in the rigorous transcriptome activation and subsequent maturation of GC lineages during PFA and depicted the gene expression signatures for each GC type.

## WNT-dependent and independent gene regulatory network during the maturation of GCs

To further characterize differences and similarities between WNT-signaling dependent and independent GC transcriptome activation during PFA, we conducted DEG analysis using the

singleCellHaystack method [23]. This approach allowed us to identify broader patterns of gene expression changes without relying on the explicit clustering of cells. Based on the expression kinetics of the top 1,000 genes of the 4,156 DEGs detected, our analysis identified four distinct DEG clusters (designated as clusters 1–4) after careful evaluation (Fig 4A and S4 Table). When clusterings with five or more groups were tested, they contained additional clusters with subtle differences in expression kinetics (S5 Fig). Thus, we inferred that four clusters effectively represented the overview of the heterogeneity of expression kinetics. Clusters 1 and 2 showed similar expression kinetics between control and *Wls* cKO GCs, with the former and the latter genes being up- and down-regulated, respectively, after the pre-GC activation (Fig 4A). Interestingly, these likely WNT-signaling independent DEGs account for 81% of the highly DEGs (cluster 1: 536 genes, cluster 2: 274 genes). In contrast, clusters 3 and 4 had fewer DEGs (cluster 3: 101 genes, and cluster 4: 89 genes), but showed contrasting regulation between control and *Wls* cKO GCs in the broad developmental stage of GCs (Fig 4A). These results suggest that, even in *Wls* cKO GCs, the majority of the transcriptome profiles associated with GC activation are still preserved. However, the absence of autocrine WNT-signaling interferes with essential pathways crucial for proper GC development.

To elucidate the functional significance of the transcriptional profiles generated through the singleCellHaystack-based clustering, we performed an enrichment analysis (Fig 4B and S5 Table). That analysis revealed a significant upregulation of pathways associated with GC activation, which were either unaffected or affected by WNT signaling. Post-activation GCs (cluster 1) included GO terms linked to active protein synthesis, regardless of the presence or absence of WNT signaling. In pre-GCs (cluster 2), genes associated with cell-cell adhesion are highly expressed, which suggests that the maintenance of plasma membrane integrity is important for their function. Our data also provide additional support for the importance of ECM production by pre-GCs, which reinforces the notion that a rich ECM surrounding primordial follicles is crucial for maintaining oocyte dormancy [13]. *Wls* cKO GC lineage (cluster 3) contains GO terms such as "transmembrane receptor protein serine/threonine kinase signaling pathway" and "positive regulation of cell differentiation". These up-regulated genes included those encoding proteins related to activation of the TGFβ/BMP pathway (*Bmp2*, *Bmp4* and *TGFB1l1*) (S5 Table). The inappropriate intercellular signaling may lead to instability or incorrect cell lineage identity. Remarkably, in the control GC lineage (cluster 4), genes associated with the maintenance of proteostasis and response to steroid hormone stimulus were up-regulated, which appears to be a hallmark of functional GCs that require WNT signaling.

Next, we investigated regulators of the pre-GC to GC transition-associated changes in gene expression. We conducted gene co-expression analysis using GENIE3 [24] with the top 1,000 DEGs identified through singleCellHaystack analysis. That analysis revealed core hub genes in GCs and their expression kinetics during PFA in relation to WNT signaling (Fig 4C and 4D). *Nr5a2*, *Tcf7* and *Tox2*, associated with cluster 1, were expressed in both *Wls* cKO and control GCs (Fig 4D). Those genes have been reported to play critical roles in GC function and fate determination [25–27]. *Ets1* and *Irx3*, which are enriched in cluster 2, were highly expressed in pre-GCs (Fig 4D). The expression patterns of these two genes are consistent between control and *Wls* cKO GCs, showing high expression before activation. *Ets1* has been implicated in the pathogenesis of polycystic ovary syndrome [28, 29], while *Irx3* has been reported to be essential for oocyte quality control through ECM production [30, 31]. *Tsc22d1* and *Bmp2*, which are upregulated in cluster 3 of *Wls* cKO GCs, have been linked to the TGFβ/BMP pathway, potentially affecting the failure of transition from pre-GC to GC. *Fosb* is exclusively expressed in control GCs like cluster 4 (Fig 4D). The expression of *Fosb* has been documented in GCs and in luteal cells, suggesting its involvement in GC differentiation [32, 33]. To further validate these transcriptional findings, we performed qPCR analysis on *Fosb*, *Bmp2*, and

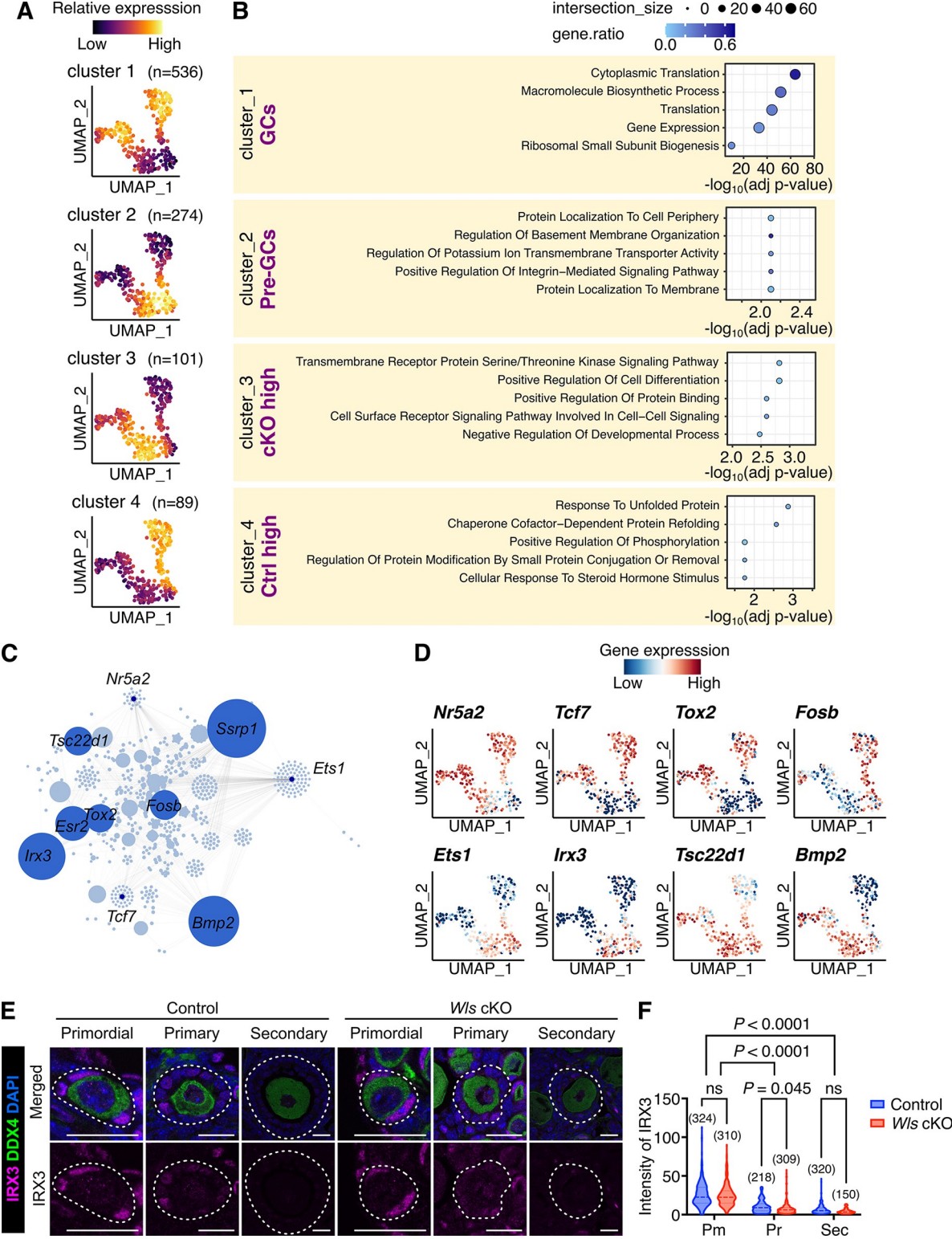

**Fig 4. Similarities and differences in gene regulatory dynamics in GC activation between control and *Wls* cKO mice.** (**A**) UMAP plot showing the average expression level of the genes in each cluster defined by hierarchical clustering (cutting into $k = 4$) of DEGs identified by the singleCellHaystack method. (**B**) Representative enriched GO terms of the clusters. (**C**) Regulatory networks visualizing potential key transcriptional regulators involved in GC differentiation. The top 10 nodes are colored dark blue. (**D**) UMAP plots showing expression levels of selected potential hub genes (*Nr5a2*, *Tcf7*, *Tox2*, *Fosb*, *Ets1*, *Irx3*, *Tsc22d1* and *Bmp2*). The color scale represents the gene expression

level scaled by the Z-score. (**E**) Immunofluorescence staining of IRX3 (magenta) and DDX4 (green) in the ovaries of 3-week-old *Wls* cKO mice and littermate control mice. Nuclei were counterstained with DAPI (blue). White dotted lines indicate follicles. Scale bars, 20 μm. (**F**) Violin plots showing fluorescence intensity of IRX3 in (pre-)GCs determined from images similar to those in (E). Horizontal lines represent the median. Statistical comparison was performed by two-way ANOVA with Sidak's multiple comparisons test; ns denotes not significant. Pm, primordial; Pr, primary; Sec, secondary follicles. Sample numbers are indicated in parentheses.

*Tsc22d1* using cDNA from the same samples analyzed in our scRNA-seq data (S6A and S6B Fig). The qPCR results confirmed the differential expression patterns observed in the sequencing data, showing upregulation of *Bmp2* and *Tsc22d1* in *Wls* cKO GCs and increased *Fosb* expression in control GCs. These findings strengthen the link between WNT signaling and the regulation of these genes. Immunostaining for the inferred hub regulator IRX3 validated the depicted expression kinetics, showing higher expression in pre-GCs that decreased during the maturation process, with no notable difference between control and *Wls* cKO groups (Fig 4E and 4F). Taken together, our analysis characterized the molecular signature of unexpected similarities and key differences in the landscape of GC transcriptome activation during PFA between the presence or absence of WNT signaling and identified their possible core regulators.

## Transcriptomic consequences in oocytes that are impacted by disrupted WNT signaling and GC mis-differentiation

We then investigated the impact of disrupted WNT signaling on GC maturation on the oocyte transcriptome during PFA. We conducted a subpopulation analysis by integrating data from both the control (Oocyte in Fig 2A) and *Wls* cKO (Oocyte in Fig 2B) datasets, excluding the altered oocyte cluster from the analysis. The clustering analysis with Seurat identified three distinct subclusters inferred as O1; Dormant, O2; Activated-Ctrl, and O3; Activated-cKO (Fig 5A–5D). These subclusters were annotated according to the known increased expression patterns of maternal genes associated with PFA, such as *Zp1*, *Zp2*, *Zp3* and *Gdf9* (Fig 5E). The dormant cluster consisted of both control and *Wls* cKO mice-derived oocytes (Fig 5B). Upon oocyte activation, we found a subtle change in gene expression patterns between the control and *Wls* cKO samples, detecting a small number of DEGs that characterized each sub-cell type (Dormant (O1): 61, Activated-Ctrl (O2): 316, Activated-cKO (O3): 10) (S6 Table).

Prior to a detailed examination of these subclusters, we outlined the features of the excluded altered oocyte cluster. Principal component analysis (PCA) across all oocyte samples revealed a representation of altered oocyte characteristics in PC1, whereas PC2 appeared to represent oocyte growth (S7A Fig). Analysis of developmental oocyte marker expression and flow cytometry-derived cell size data indicated a mixed composition of dormant and activated oocytes within the altered oocyte subcluster (S7B Fig). The significantly low values for total sequence and percent intron, referred to in Fig 2E, suggest transcriptional inactivity in the altered oocytes (S7C Fig). Factor loading in the PCA suggests that these distinct characteristics result from a widespread reduction in gene expression, including *Mdm4*, known for its anti-apoptotic role (S7D and S7E Fig). While our data did not provide insights into the reasons behind the transcription repression, the altered oocytes were presumed to be weakened cells, warranting their exclusion to focus on the transcriptomic changes associated with PFA.

DEGs for the Dormant (O1) subcluster encompass *Ankrd36*, whose mutation has been implicated in diminished ovarian reserve [34], as well as *Pm20d1*, which has been noted for its potential involvement in the pathogenesis of polycystic ovary syndrome [35, 36] (Fig 5D). DEGs for the Activated-Ctrl (O2) subcluster contain famous oocyte-specific genes typically found in mature oocytes like *Dppa5a*, *Kpna7*, *Khdc3* and *Nlrp4a* [37–40], whereas the

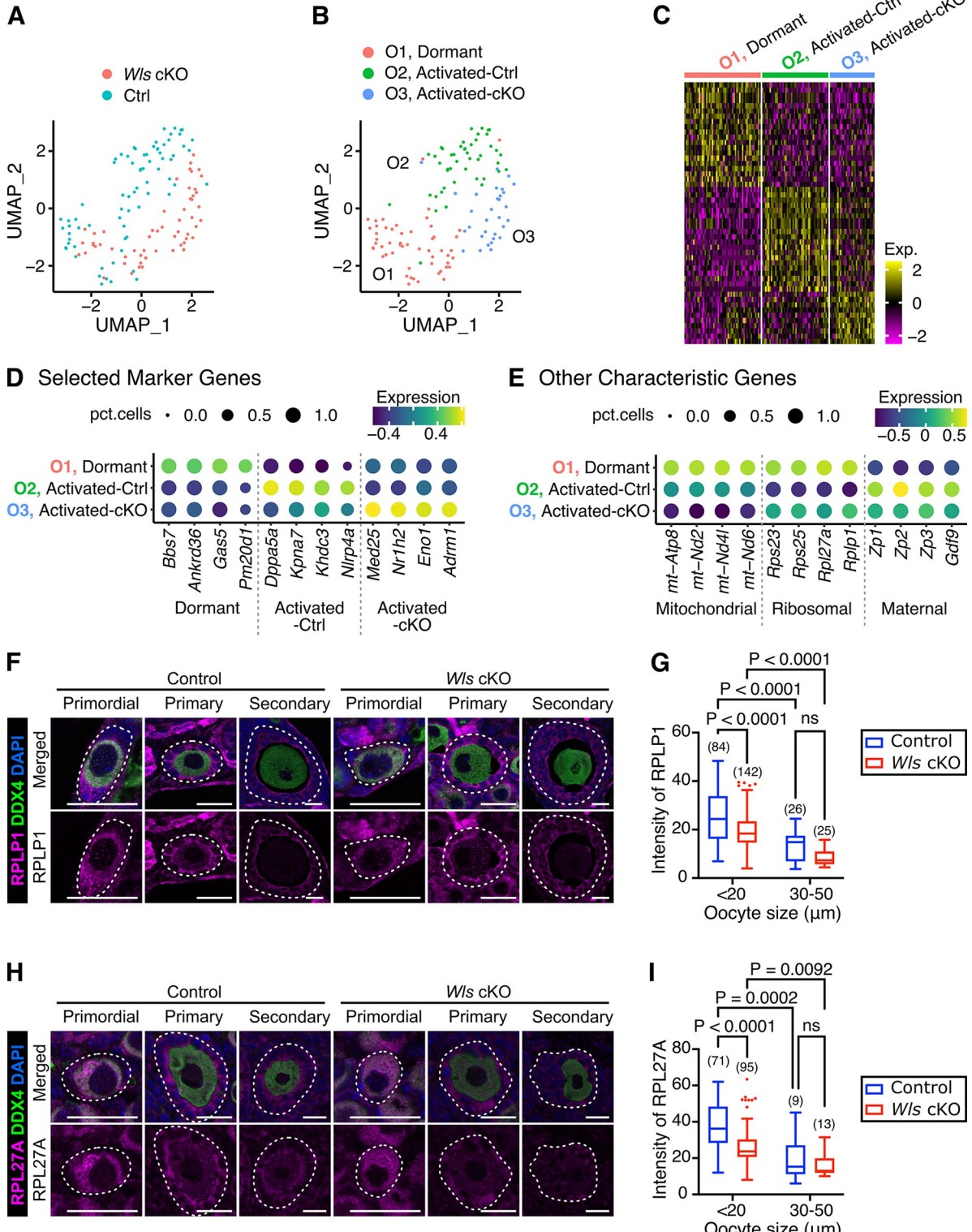

**Fig 5. Distinct gene expression patterns in dormant and in activated oocytes under the influence of insufficient GC maturation.** (A) UMAP plot of oocytes color-coded by genotype (Ctrl, blue; *Wls* cKO, red). (B) UMAP plot showing the different cell subclusters of oocytes. (C) Heatmap of the top 20 DEGs of each subcluster; for the activated-cKO subcluster, all 10 DEGs were used. (D, E) Dot plots showing the expression of selected markers for subclusters or characteristic features. The dot size represents the ratio of cells expressing a specific marker (TPM > 1), while the color indicates the average gene expression level scaled by the Z-score. (F) Immunofluorescence staining of RPLP1

(magenta) and DDX4 (green) in the ovaries of 3-week-old *Wls* cKO mice and littermate control mice. Nuclei were counterstained with DAPI (blue). White dotted lines indicate follicles. Scale bars, 20 μm. (H) Immunofluorescence staining of RPL27A (magenta) and DDX4 (green) in the ovaries of *Wls* cKO mice and littermate control mice at PD7. Nuclei were counterstained with DAPI (blue). White dotted lines indicate follicles. Scale bars, 20 μm. (G, I) Boxplots depicting fluorescence intensity of RPLP1 (G) or RPL27A (H) in oocytes determined from immunofluorescent images. Sample numbers are indicated in parentheses. Statistical comparison was performed by two-way ANOVA with Sidak's multiple comparisons test; ns denotes not significant.

Activated-cKO (O3) subcluster showed lower expression of these genes (Fig 5D). Among the 10 DEGs of the Activated-cKO (O3) subcluster, genes such as *Nr1h2*, which has been implicated in lipid metabolism, *Eno1*, linked to glycolysis, and *Adrm1*, a constituent of proteasomes, imply a shift in trophic status (Fig 5D).

Due to the limited number of DEGs, GO analysis yielded only a few significant terms (S7 Table). Yet, we observed high expression levels of mitochondrial genes and ribosome-related genes in the Dormant (O1) subcluster (Fig 5E), which have been documented in previous studies [41]. Immunostaining verified the expression levels of RPLP1 and RPL27A in DDX4-positive oocytes (Fig 5F and 5H). Both RPLP1 and RPL27A showed the highest expression in oocytes smaller than 20 μm, which are presumed to be dormant in primordial follicles (Fig 5G and 5I), while downregulated expression of RPLP1 and RPL27A was noted in oocytes 30–50 μm diameter, which are often observed in secondary follicles. In oocytes smaller than 20 μm, *Wls* cKO mice showed a significantly lower intensity of these ribosomal proteins compared to controls, although the difference was not substantial.

We then examined DEGs using the singleCellHaystack method to test how GC mis-differentiation in *Wls* cKO ovaries affects the transcriptome of interacting oocytes. That analysis identified 1,132 DEGs that were effectively represented by four expression patterns (clusters 1–4) (Fig 6A and S8 Table). We expected that the retained nuclear localization of FOXO3 in growing *Wls* cKO oocytes would result in the constant up-regulation of "dormant" genes even after PFA stimulus. However, we failed to detect a cluster of genes expressed in such a manner (Fig 6B). Instead, the majority of genes (83%) showed similar expression kinetics between control and *Wls* cKO samples with associated expression changes with PFA (Cluster 1: 224 genes up-regulated in dormant oocytes; Clusters 3 and 4: 429 and 281 genes, respectively, up-regulated in activated oocytes) (Fig 6A). Interestingly, Cluster 2 (198 genes) showed a significant down-regulation both in dormant and in activated oocytes in *Wls* cKO mice (Fig 6B). We then explored the functional association of these DEGs using available databases (Gene Ontology, UK Biobank GWAS and MSigDB Oncogenic Signature). Although there were no significantly enriched terms from those databases, there were notable possible WNT-associated terms of "Wnt up-regulation associated oncogenes" (*Btg4*, *Fabp5*, *Myo5b*, *Mtf1*, *Ckap4*) and "age at menopause" (*Ddx17*, *Akap13*, *Slco4a1*) at the top of the term lists, suggesting a possible molecular crosstalk between (pre-)GCs and oocytes related to the cause of infertility via defective WNT-signaling (S9 Table). After examining all available TF databases implemented in Enrichr, there was no evidence of enrichment of FOXO3-related terms in any of the DEG clusters, even when the kinetics were classified into 4 to 6 clusters (S8 Fig).

Finally, we investigated gene regulatory networks during PFA by performing gene co-expression analysis based on the DEGs detected in the Seurat analysis (Fig 6C). We identified hub genes within the top 20, whose expression was well represented before and after activation and by genotype (Fig 6D). *Zfp57* and *Foxr1*, known as oocyte-specific genes, are hub gene candidates that are preferentially expressed in activated oocytes, including the Activated-Ctrl (O2) and Activated-cKO (O3) subclusters (Fig 6D). Consistent with our results, *Foxr1* has been reported to be upregulated after PFA [42], suggesting that these transcription factors are likely activated during PFA regardless of aberrant GC cell activation in *Wls* cKO mice. *Nr1h2* and

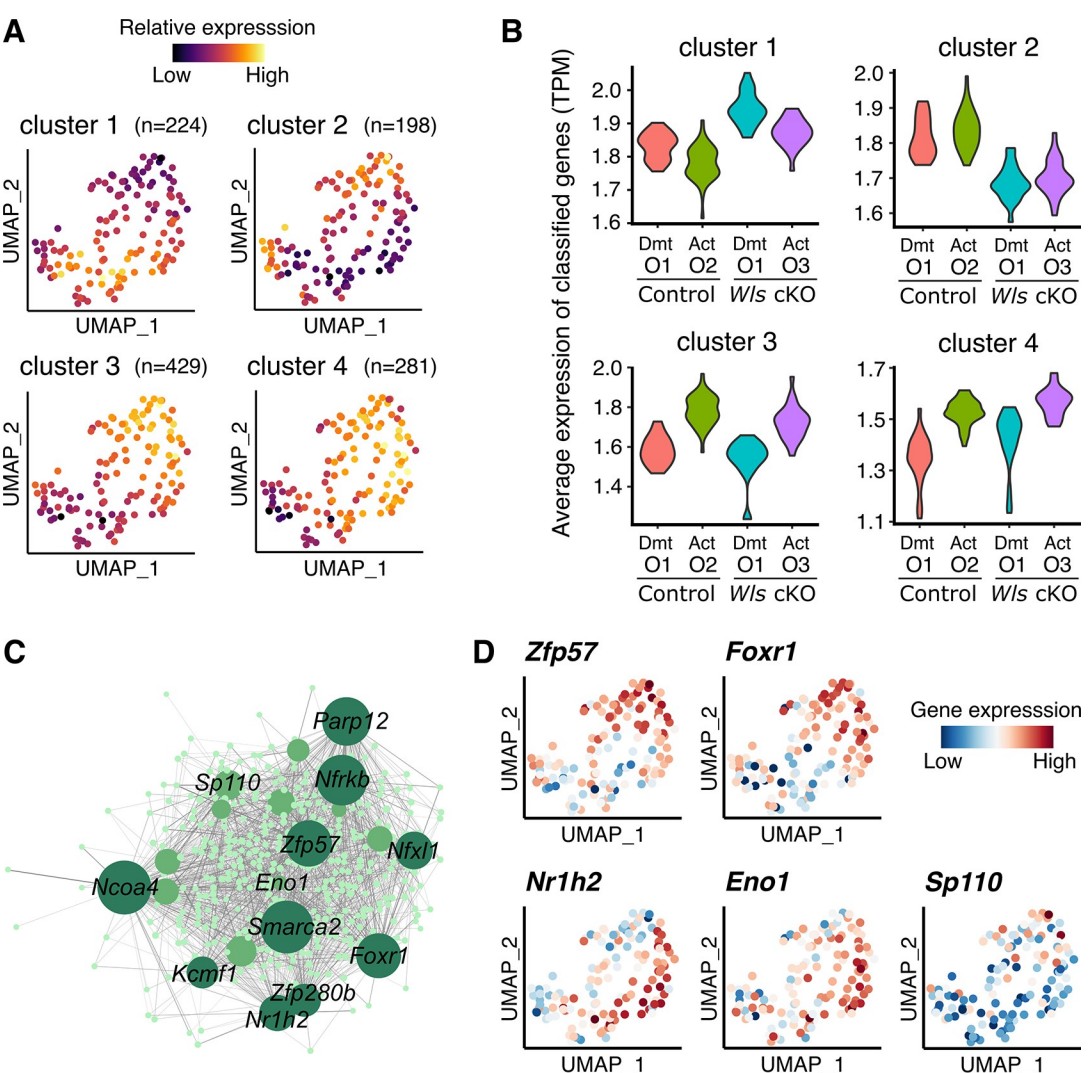

**Fig 6. Analysis of possible molecular crosstalk and regulatory features in the absence of WNT signaling.** (**A**) UMAP plots depict the average expression level of DEGs in each cluster defined by hierarchical clustering (cutting into $k = 4$) identified by the singleCellHaystack method. (**B**) Violin plots indicating the expression of DEGs for each cluster within oocyte subclusters (O1–O3) from control and from *Wls* cKO samples. Dmt, Dormant. Act, activated. (**C**) Regulatory networks visualizing potential key transcriptional regulators in oocyte activation. The top 10 nodes are colored dark green, nodes ranked 11–20 are colored green, and the others are colored light green. (**D**) UMAP plots showing expression levels of selected potential hub genes (*Zfp57*, *Foxr1*, *Nr1h2*, *Eno1* and *Sp110*). The color scale represents the gene expression level scaled by the Z-score.

*Eno1*, while DEGs in the O3 subcluster are hub gene candidates upregulated in oocytes from *Wls* cKO mice (Fig 6D). To validate the reproducibility of our scRNA-seq data, we conducted qPCR analysis of *Nr1h2* and *Eno1*, which confirmed their upregulation in the O3 subcluster (S6C and S6D Fig). *Nr1h2* is known to be predominantly expressed in non-growing oocytes, indicating an unbalanced undifferentiated state in *Wls* cKO ovaries [43]. Another hub gene candidate, *Sp110*, is expressed predominantly in the Activated-Ctrl (O2) subgroup and is known to increase after PFA (Fig 6D) [42]. *Sp110* expression might be indirectly regulated by GCs receiving WNT signaling. Altogether, these results depict regulatory features of oocyte gene expression that are affected by the direct and indirect impact of surrounding GCs with or without WNT signaling during PFA.

## Discussion

In this study, we uncover for the first time the single-cell resolution regulatory dynamics of a PFA process using a unique model, *Wls* cKO mice. The purpose of this paper includes addressing three main questions. First, we asked: (i) What are the molecular mechanisms by which cKO of *Wls* causes defective pre-GC to GC development? In *Wls* cKO mice, nearly all GCs, except pre-GCs, have a disorganized phenotype, which results in complete female infertility due to the inability to support oocyte growth. Consistent with that phenotype, the transcriptome landscapes of control and *Wls* cKO cells were similar until the pre-GC stage prior to PFA. The transcriptomic features of *Wls* cKO mice became apparent immediately in GCs after PFA, although the features of *Wls* cKO-derived oocytes were more modest than expected at the transcriptomic level.

In the GC lineage from *Wls* cKO ovaries, some genes were erroneously up-regulated after the initiation of PFA. Particularly, TGFβ/BMP signaling appears to remain active throughout GC subpopulations, marked by the elevated expression of *Bmp2*. TGFβ superfamily members are expressed in various ovarian cell types [44]. While *Bmp2* typically functions in late GCs of antral follicles during late folliculogenesis [45, 46], the elevated expression of *Bmp2* in atretic GCs during early folliculogenesis in rats [44] suggests a potential for atretic conditions in GCs of *Wls* cKO mice. A similar abnormal morphology of GCs in *Wls* cKO mice has been reported in GCs under Müllerian inhibiting substance (MIS)-treatment, which suppresses the proliferation and differentiation of mouse GCs [17]. Both MIS signaling and WNT pathways play integral roles in GC proliferation and maturation. Given the identified potential upregulation of TGFβ/BMP signaling both in MIS-treated mice and in *Wls* cKO mice, the precise suppression of TGFβ/BMP signaling may be crucial for the pre-GC to GC transition.

Notably, genes associated with protein quality control have a decreased expression in *Wls* cKO GCs, indicating a potential dysfunction of proteostasis. This result supports the potential regulation of protein quality control by WNT signaling, suggesting there is a crucial interplay to achieve cellular homeostasis and function. In addition to that, the observed downregulation of *Fosb* in *Wls* cKO GCs as a potential expression regulator, is reminiscent of its reported decline with age in primate datasets [41]. This suggests intriguing parallels between the molecular profiles of *Wls* cKO GCs and those associated with aging. Unraveling the intricate connections between WNT signaling and the aging process in GCs may provide insights into the age-related decline in ovarian function.

Based on the interaction between GCs and oocytes, we next asked: (ii) How does GC misdifferentiation in *Wls* cKO mice affect the transcriptome of interacting oocytes? In this study, we identified an altered oocyte cluster characterized by a low transcriptional level in *Wls* cKO mice, which allows us to elucidate the impact of GC dysfunction on oocyte conditions. That cluster appeared to encompass both dormant and activated oocytes, suggesting a complex interplay between GC maturation and oocyte state. The present data set does not conclusively elucidate the reason(s) for the division of the oocyte population into two distinct states in *Wls* cKO ovaries. However, based on the presence of the altered oocyte cluster and the terminal phenotype of oocyte depletion in adult *Wls* cKO ovaries [15], we infer that all oocytes will eventually shift to the altered oocyte cluster and undergo atresia. Further clarification of the cellular lineage and the impact of unhealthy GCs may offer valuable insights into the underlying mechanisms governing oocyte integrity and atresia.

Activated oocytes in *Wls* cKO mice showed modest transcriptome alterations compared to the control, with parallel expression changes associated with PFA. However, the singleCell-Haystack method revealed a notable down-regulation of the set of genes even before PFA initiation in *Wls* cKO mice-derived oocytes, which contrasts with pre-GCs in *Wls* cKO ovaries that

appeared normal in the transcriptome as well as our previous histological observations [15]. The enriched tendency of genes known to have a positive regulatory correlation with WNT only in control oocytes suggests that these down-regulated genes are caused by WNT signaling-dependent molecular cross-talk in (pre-)GC-oocyte interactions, rather than being solely influenced by the surrounding microenvironments. This result implies the possibility that oocytes directly receive WNT ligands from (pre-)GCs. Previous reports of the over-activation of WNT signaling in oocytes by the constitutive expression of active β-catenin showed no phenotype in folliculogenesis. It may be that gain-of-function experiments do not adequately reveal the role of WNT signaling in oocyte growth. However, the non-significant term and the detection of only a few DEGs between subclusters support the idea that the absence of the proper microenvironment, which is facilitated by appropriately differentiated GCs, is likely the critical cause of atretic oocytes observed in *Wls* cKO mice at a later stage of folliculogenesis [15].

Observing the unsynchronized PFA process in oocytes, we asked: (iii) What is the molecular signature of this process in oocytes that are growing despite the possible indication of retained dormancy? Although the oocytes of *Wls* cKO mice are activated and develop to a certain degree, we previously reported that these growing oocytes partially maintain dormant characteristics, as indicated by the nuclear localization of FOXO3 [15]. In other words, oocyte activation at PFA encompasses two processes: an increase in size and a break from dormancy. *Wls* cKO mice serve as a model where only the growth process occurs. In this study, the PFA process was described as a transcriptomic change. We found that dormancy in oocytes has weak associations with the transcriptional landscape in this context. Nuclear FOXO3 may be insufficient or uncorrelated to the arrest in oocyte size growth and transcriptome activation, key processes in oocyte activation. However, it remains possible that the oocytes we previously observed in *Wls* cKO ovaries belonged to the altered oocyte cluster and that extensive transcriptional downregulation prevented us from detecting transcriptomic changes. WNT-mediated GC function likely facilitates FOXO3 nuclear export in oocytes, but Kit signal components, which are upstream of FOXO3 phosphorylation, are unexpectedly up-regulated in *Wls* cKO ovaries [15]. These results suggest a potential compensatory feedback mechanism and/or the importance of an environment such as nutrition and mechanical stress provided by surrounding GCs, emphasizing the need for further experimental studies to fully understand the intricate interactions in this regulatory network.

In conclusion, our study on *Wls* cKO ovaries uncovered the intricate impact of WNT signaling on the dynamics of GC differentiation and oocyte activation. WNT signaling may impact reproductive performance through its multifaceted roles in sexual differentiation, follicle development, luteinization and steroid production [47, 48]. Upregulated WNT signaling is linked to ovarian cancer in mice and in humans [49, 50], and altered WNT-related gene expression has been reported in the ovaries of polycystic ovary syndrome, a potential cause of infertility [51]. Our results contribute to the growing knowledge in this area and may serve as a valuable resource for future investigations into the clinical relevance of WNT signaling. Further studies dissecting the impact of aberrant GCs should provide insights into reproductive biology to address infertility.

## Materials and methods

### Animals

*Sf1-Cre* mice (stock no. 012462), *Wls*flox mice (stock no. 012888), *Ddx4-Cre* mice (stock no. 006954) and *Ai9* mice (stock no. 007909) were obtained from The Jackson Laboratory [52–55]. *Wls*del mice with a ubiquitous *Wls*flox allele deletion were obtained by crossing *Wls*flox mice with *Ddx4-Cre* mice. Littermate controls, either *Sf1-Cre;Wls*flox/+ or *Wls*flox/del, were

utilized for immunostaining comparisons with *Sf1-Cre;Wls*<sup>flox/del</sup>(*Wls* cKO). *Sf1-Cre;Ai9* mice were used to confirm the efficiency of Cre. To prepare single cells for RNA sequencing, *Sf1-Cre;Wls*<sup>flox/+</sup>*;Ai9* mice served as controls for *Sf1-Cre;Wls*<sup>flox/del</sup>*;Ai9* (*Wls* cKO) mice. All mouse lines were maintained on a mixed genetic background.

Mice were housed in a temperature- and humidity-controlled environment with a 12-hour light/dark cycle, and were provided with food and water ad libitum. Euthanasia was performed via cervical dislocation. No anesthesia or analgesia was required during euthanasia as the method was quick and performed by trained personnel to ensure minimal suffering. All tissues were collected after euthanasia. Efforts were made to minimize animal distress throughout the study. All animal experiments were approved by the Institutional Animal Care and Use Committee of RIKEN (approval number: A2017-13).

## Immunostaining and histology

Immunofluorescence staining was performed using ovaries that had been fixed overnight at 4˚C with 4% paraformaldehyde in phosphate-buffered saline (PBS). The fixed tissue was dehydrated, embedded in paraffin and then sectioned at a thickness of 5 μm. The sections were depleted of paraffin and rehydrated according to standard protocols. For antigen retrieval, the sections were incubated either at 110˚C for 15 min with citrate buffer (pH 6.0) or with Tris-EDTA buffer (pH 9.0). After washing with PBS containing 0.1% Tween 20 (PBST), the sections were incubated for 1 h at room temperature in a blocking buffer, stained overnight at 4˚C with primary antibodies, and then exposed for 2 h at room temperature to a 1:500 dilution of appropriate secondary antibodies labeled with Alexa Fluor 488, 568 or 647 (A11057, A21206 and A10042, Thermo Fisher Scientific; or 715-545-151, Jackson Immuno Research). The primary antibodies used included mouse monoclonal anti-DDX4 (1:300; ab27591, Abcam) rabbit polyclonal anti-DDX4 (1:300; ab13840, Abcam), goat polyclonal anti-PDLIM4 (1:133; NB100-1382, Novus Biologicals), rabbit monoclonal anti-IRX3 (1:100, ab242133, Abcam), rabbit polyclonal anti-RPLP1 (1:200; NBP1-81293, Novus Biologicals) and anti-RPL27A (1:150; NBP2-38025, Novus Biologicals). Antibodies were diluted in blocking buffer or Can Get Signal immunostain solution (NKB-401, Toyobo). DNA was counterstained with 4′,6-diamidino-2-phenylindole (DAPI). Samples were mounted with VECTASHIELD Vibrance Antifade Mounting Medium (H-1700, Vector Laboratories).

To assess Cre recombination efficiency, ovaries were fixed overnight at 4˚C with 4% paraformaldehyde in PBS. The fixed tissues were immersed in sucrose gradients (10%, 20% and 30%) in PBS sequentially at 4˚C; after which the tissues were embedded in an optimal cutting temperature compound. Frozen samples were sectioned at 6 μm using a CryoStar NX70 cryotome (Leica Microsystems). Ovarian cryosections from *Sf1-Cre;Ai9* mice were incubated in blocking buffer for 1 h at room temperature, stained overnight at 4˚C with rabbit polyclonal anti-DDX4 (1:500; ab13840; Abcam) antibody, followed by incubation with donkey anti-rabbit IgG Alexa Fluor 488 (A21206, Thermo Fisher Scientific). DNA was counterstained with DAPI. Samples were mounted using VECTASHIELD Vibrance Antifade Mounting Medium (Vector Laboratories).

PAS-H staining was performed according to a standard protocol. In brief, ovaries were fixed in Bouin's solution, embedded in paraffin, and sectioned at a thickness of 5 μm. The sections were hydrated and treated first with a periodic acid solution for 10 min and then with Schiff's reagent for 15 min. Nuclei were counterstained with hematoxylin.

## Ovary dissociation and single-cell sorting

At postnatal day (PD) 7, ovaries from euthanized control mice (n = 3) and from *Wls* cKO mice (n = 8) were collected. Ovaries were treated with a CTK solution (consisting of 0.25% trypsin

(Thermo), 0.1% collagenase type IV (Gibco), 20% KSR (Invitrogen) and 1 mM $CaCl_2$ in PBS) for 30 min at 37˚C, then further treated with Accutase (Nacalai) at 37˚C for 15 min. The reaction was halted by adding L-15 medium with 0.1% penicillin-streptomycin and 5% FBS, and the cells were dissociated by pipetting. The cell suspension was filtered through 70 μm cell strainers, followed by centrifugation at 400 g for 5 min and resuspension in sorting buffer (7.5% BSA (Life Technologies) in PBS). Blocking was performed with rat IgG (5 mg/mL, I8015, Sigma-Aldrich) for 15 min on ice. Subsequently, cell surface staining was conducted using anti-ckit-APC (1:50, 105812, BioLegend) and anti-CD45-Alexa Fluor 488 (1:200, 103122, BioLegend) for 15 min on ice. After centrifugation at 400 g for 5 min, cells were resuspended in sorting buffer. Prior to single-cell sorting by flow cytometry, samples were passed through 40 μm cell strainers, and DAPI (1:10,000) was added. Control and *Wls* cKO samples were each sorted using MoFlo Astrios EQ with Summit v 6.3.0. software (Beckman Coulter) into 384-well plates containing 286 tdTomato-positive somatic cells and 95 ckit-positive oocytes. DAPI-positive dead cells and CD45-positive leukocytes were excluded.

## RamDA-seq library preparation

Library preparation for RamDA-seq was performed according to a previously report [19]. In the RT-RamDA reaction, we added 1:5,000,000 ERCC RNA Spike-In Mix I (Thermo Fisher). However, to reduce reagent costs and rRNA-derived sequencing reads, the method was modified as follows: To suppress rRNA-derived cDNA synthesis, 0.008 units/μl XRN-1 (NEB) was added to the genomic DNA digestion mix, and the reaction was performed at 30˚C for 5 min. To reduce reagent costs, a liquid dispenser Mantis (Formulatrix) was used to perform the reaction up to second strand synthesis at 1/10 the reaction volume. After that, 2.5 μl 0.8 X Tagment DNA Buffer (Illumina) diluted with RNase-free water was added to 0.5 μl of the second strand synthesis products for dilution. Then, 0.5 μl Amplicon Tagment Mix (Illumina) was added to 1.5 μl of the diluted product, and tagmentation was performed at 55˚C for 10 min. After that, the library was prepared using 1/10 the reaction volume according to the Nextera XT DNA Library Prep Kit (Illumina) procedure. Twelve PCR cycles were performed. Somatic and oocyte library products were collected separately in equal volumes of 1 μl/well. Subsequently, 50 μl of each sample was fractionated from the equal mixed library, purified with 1.2X AMPureXP (Beckman Coulter), and eluted with 25 μl TE. To prevent large discrepancies in the number of sequencing reads per cell between somatic and oocyte cells, purified libraries were mixed to ensure a constant average molar per cell type using the quantitative results from the Bioanalyzer Agilent High-Sensitivity DNA Kit and the ratio of the number of cells sorted (somatic cells: oocytes = 286:95). The pooled library of 2.3 pM was sequenced in single end read 76 cycles using NextSeq 500 High Output Kit v2.5.

## Quality control metrics for RamDA-seq data

The RamDA-seq reads were mapped to the mouse genome (GRCm38) using Hisat2 v2.2.0 [56] after trimming adaptor sequences and low quality bases using Fastq-mcf v1.1.2 in the ea-utils package [57]. The resulting binary alignment/map (BAM) files were sorted using samtools [58] and the final mapping quality was assessed using RSeQC v3.0.1 [59]. The gene count table was obtained using the featureCounts v2.0.1 tool in the Subread package [60] with the option '-t exon -g gene_id' and the annotation gtf file from GENCODE (vM25). We evaluated the quality control metrics and filtered out low quality samples based on the RNA integrity assessed by coverage profile along the gene body, the number of detected genes $\leq 8,000$, and outlier cells in Uniform Manifold Approximation and Projection (UMAP) clustering. In addition, the extent of sample preparation artifacts (batch effects) was investigated by exploring

genotype-associated clusters in Principal Component Analysis (PCA) and UMAP using the total gene expression and genotype-independent measures of ERCC RNA spike-in transcripts, respectively. We performed downstream analysis on the remaining data from 350 and 371 cells from control and *Wls* cKO mice, respectively.

## Sequence data analysis

Seurat v3.2.2 [61] was used for clustering and differential gene expression analysis of scRNA-seq data for control and *Wls* cKO samples independently using the same parameters. The read counts were normalized as log scaled transcripts per million (TPM). Dimensionality reduction and clustering were performed using the 'FindClusters' function of Seurat with the default parameters except for 'resolution = 1', using the top 2,000 variable genes. Since our clustering detected a possible altered GC population, the percentage of intron reads and the abundance of nuclear ncRNA, using Malat1 as a marker, were evaluated to assess cell viability. Enrichment of GO (Biological Process) for marker genes of each cluster was tested using Enrichr [62].

Detailed subpopulation analyses were performed for each oocyte and GC population after pooling the control and *Wls* cKO datasets. In this analysis, we excluded the altered GC population in *Wls* cKO samples. We used Seurat for clustering and exploration of DEGs with the same settings used for the above whole cell dataset. DEGs, defined as false discovery rate (FDR) < 0.05, were identified using the 'FindAllMarkers' function of the Seurat. For the subpopulation analysis, due to the complicated nature of cell clustering relationships, DEGs were further explored using singleCellHaystack v0.3.4, which identifies DEGs without relying on the clustering of cells using the Kullback-Leibler Divergence method [23]. This analysis was performed using the "highD" method with the following options: for GCs, all 40 principal components with grid.points = 30, while for oocytes, top 22 principal components, whose cumulative contribution ratio just exceeds 70%, with grid.points = 10 to increase the detectability of DEGs in fewer numbers of cells. The detected DEGs were then classified into four seemingly parsimonious expression kinetic patterns after testing up to six patterns: for GCs, the top 1,000 confidential DEGs based on the FDR were used. These DEGs were used for the subsequent analysis. Enriched functional features of DEGs were tested with Enricher using available databases (Gene Ontology, UK Biobank GWAS, MSigDB Oncogenic Signature and all TF databases). To identify potentially important transcriptional regulators, regulatory network analysis was performed using the GENIE3 package, a random forest method that calculates the links between each gene and all other genes [24]. We used TPM-scaled read counts of DEGs (oocytes: FDR < 0.05 by Seurat; GCs: top 1,000 DEGs by singleCellHaystack) as input. With the information of known mouse transcription factors [63], only the transcriptional regulator-targets connected with thresholds above 0.07 were retained and used for the network analysis. The resulting networks were visualized using CytoScape 3.10 [64].

## Single-cell RT-qPCR validation

To validate the single-cell RNA-seq results, we performed single-cell reverse transcription-quantitative PCR (scRT-qPCR) on cDNA synthesized during the RamDA-seq protocol from previously sequenced single-cell plates. For the qPCR reactions, 1/20th of the cDNA product derived from a single cell after second-strand synthesis was used as input. We targeted five specific genes along with Gapdh as an internal control. The primer sequences for these genes are listed in S10 Table. Quantitative PCR was performed using a LightCycler 480 (Roche). Each reaction consisted of 3.5 μL of qPCR reaction mix (2.5 μL of 2x QuantiTect SYBR Green Master Mix, 0.03 μL each of forward and reverse primers at 100 μM, and 0.94 μL of 0.0015% NP40)

added to 1.5 μL of diluted cDNA in a 384-well plate. The cycling conditions included an initial activation step at 95 ˚C for 15 minutes, followed by 40 cycles of denaturation at 95 ˚C for 15 seconds and annealing, extension at 60 ˚C for 1 minute. A melting curve analysis was performed to ensure product specificity. Cq values were converted to expression levels using the formula Log2 expression = LoD (Limit of Detection) Cq–Cq [Assay] [65]. If this value was negative, the result was assigned a value of "0". The LoD was determined using a 5-fold dilution series (7 points; n = 8) of cDNA derived from 1 ng of total RNA from mouse ovaries. The mean Cq of the most diluted sample concentration detected in all n = 8 samples was taken as the LoD. The LoD for each primer set is as follows: *Nr1h2*: 30.4825412, *Eno1*: 31.6995186, *Bmp2*: 29.933429, *Tsc22d1*: 30.395342, *FosB*: 32.8733681, *Gapdh*: 32.3047353. For normalization across oocyte samples, we utilized FSC-Area, representing cell size, to account for variability in RNA content due to differences in cell maturation.

## Image analysis

Immunostaining was examined using a slide scanner (Axio Scan.Z1, Zeiss) or a confocal laser scanning microscope (LSM780, Zeiss). All images were taken with Axio Scan.Z1 utilizing the tile scan and automated stitching functions.

The thickness of the GC layer was determined as half the difference between the diameters of the follicle, and oocytes were measured in PAS-H–stained ovarian sections using ImageJ software (NIH). Oocytes with visible nuclei were analyzed in a single section per ovary from three ovaries for each genotype at PD7.

For the measurement of IRX3 signals in pre-GCs/GCs, ovarian sections were subjected to immunofluorescence staining for IRX3, as well as for the oocyte marker DDX4. Follicle boundaries were delineated based on histology, with the DAPI-positive area serving as the measurement region for (pre-)GC analysis. The fluorescence intensity of IRX3 in each region was measured using ImageJ software. More than four ovaries for each genotype, as well as more than one section per ovary, were analyzed.

For the quantification of RPLP1 or RPL27A intensities, images of ovarian sections immunostained for RPLP1 or RPL27A were analyzed using ImageJ software in each oocyte area. The regions of the oocyte were determined manually utilizing ImageJ. Oocytes with visible nuclei were analyzed in a single section per ovary from three ovaries for the control group or from four ovaries for the *Wls* cKO group.

## Statistical analysis

All statistical analyses were performed using GraphPad Prism 9 software. Tests included the nonparametric Mann-Whitney matched-pairs test, unpaired multiple t-tests with the Holm-Sidak correction, two-way analysis of variance (ANOVA) with Sidak's post-hoc test for multiple comparisons, and a chi-square test for trend for the contingency table. Regression lines were fitted with the scatterplot of GC thickness data for control and for *Wls* cKO mice, and then analysis of covariance (ANCOVA) analysis was performed. A P value of <0.05 was considered statistically significant.

## Supporting information

**S1 Fig. Optical detection of cells and quality control for single-cell RNA sequencing.**
UMAP plots showing tdTomato-expressing somatic cells (**A**), the KIT-APC positive oocyte population (**B**), the total number of mapped reads (total seq) (**C**), and the number of detected genes (number of genes) (**D**). The color scale represents the gene expression level scaled by the

Z-score.
(TIF)

**S2 Fig. Quality control of single-cell RNA sequencing data.** (**A**) Number of detected genes or (**B**) Gene body coverage before and after trimming of the single-cell RNA sequencing data, showing the normalized distribution of read coverage across the length of genes. Trimming was performed following MultiQC guidelines. (**C**) UMAP plot showing two outlier cells that were identified and excluded from the control group of somatic cells.
(TIF)

**S3 Fig. Key quality metrics of retained cells following quality control procedures.** PCA plots with overlaid key quality metrics post-QC filtering, including the assigned genome rate, mitochondrial gene mapping rate, and rRNA mapping rate.
(TIF)

**S4 Fig. Evaluation of potential batch effects.** (**A**) UMAP plots of ERCC RNA spike-in transcripts showing highly mixed distribution regardless of genotype (left) and cell type (right). (**B**) PCA plots based on expression of all genes showing the first three principal components were not associated with genotype.
(TIF)

**S5 Fig. DEG clustering dendrogram of GC populations based on expression kinetics.** Diagram of hierarchical clustering based on the kinetics of DEGs obtained by the singleCellHaystack method in the GC population. Color bars represent the clustering result when the dendrogram is cut into 4–6 clusters. The number to the left of each bar indicates the number of clusters. The average expression levels of the genes in each cluster were projected onto UMAP plots below the corresponding bars. More clusters lead to a greater subdivision of the main DEG cluster.
(TIF)

**S6 Fig. Validation of differential gene expression in *Wls* cKO and control mice during PFA.** (**A, B**) Gene expression validation in granulosa cells (GCs) measured by single-cell RT-qPCR (scRT-qPCR). The expression levels of *Fosb*, *Bmp2*, and *Tsc22d* were analyzed. (A) Violin plots depict gene expression levels across GC subclusters, while (B) UMAP projections show the spatial distribution patterns of these genes. (**C, D**) scRT-qPCR validation of gene expression in oocytes. The expression levels of *Nr1h2* and *Eno1* were analyzed, with (C) showing violin plots and (D) displaying UMAP projections to compare expression across oocyte subclusters.
(TIF)

**S7 Fig. Characteristic features of the altered-oocyte subcluster.** (**A**) PCA plots of oocytes. Different colors correspond to subclusters of oocytes. The principal components on the axes capture the major sources of gene expression variation, with their respective contributions indicated in the axis labels. (**B**) Dot plot showing the expression of growing oocyte markers. The dot size represents the ratio of cells expressing a specific marker (TPM > 1), while the color indicates the average gene expression level scaled by the Z-score. (**C**) UMAP plots showing cell size, the total number of mapped reads (total sequence), the ratio of intronic reads (percent intron), the PC1 positive factor loading and the PC1 negative factor loading. (**D**) UMAP plots showing the top 6 characterized genes of PC1 positive factor loading. (**E**) UMAP plots showing the top 6 characterized genes of PC1 negative factor loading.
(TIF)

**S8 Fig. DEG clustering dendrogram of oocyte populations based on expression kinetics similarity.** Diagram of hierarchical clustering based on the kinetics of DEGs obtained by the singleCellHaystack method in the oocyte population. Color bars represent the clustering result when the dendrogram is cut into 4–6 clusters. The number on the left of each bar indicates the number of clusters. The average expression level of the genes in each cluster were projected onto UMAP plots below the corresponding bars.
(TIF)

**S1 Table. List of top 20 DEGs for each cluster identified from the PD7 control ovary dataset.**
(XLSX)

**S2 Table. List of top 20 DEGs for each cluster identified from the PD7 *Wls* cKO ovary dataset.**
(XLSX)

**S3 Table. List of DEGs identified with Seurat from granulosa cell subpopulation dataset.**
(XLSX)

**S4 Table. List of DEGs identified with the singleCellHayStack from the granulosa cell subpopulation dataset.**
(XLSX)

**S5 Table. List of significantly enriched GOs for cluster (K = 4) of DEGs identified by the singlecellHayStack from the granulosa cell subpopulation dataset.**
(XLSX)

**S6 Table. List of DEGs identified with Seurat from oocyte subpopulation dataset.**
(XLSX)

**S7 Table. List of enriched GOs based on DEGs identified with Seurat from oocyte subpopulation dataset.**
(XLSX)

**S8 Table. List of DEGs identified with the singleCellHayStack from the oocyte subpopulation dataset.**
(XLSX)

**S9 Table. Disease database showing ovarian or WNT related defects were at top of the enrichment table.** Top 10 terms were listed for the two databases.
(XLSX)

**S10 Table. List of primer sequences for scRT-qPCR analysis.**
(XLSX)

## Acknowledgments

We thank A. Matsushima (RIKEN BDR) for assistance with the infrastructure for the data analysis. This research was supported by the Medical Research Center Initiative for High Depth Omics, Nanken-Kyoten, and Single-cell Omics Laboratory in TMDU.

## Author Contributions

**Conceptualization:** Hinako M. Takase, Itoshi Nikaido.

**Data curation:** Mika Yoshimura.

**Formal analysis:** Tappei Mishina, Tetsutaro Hayashi, Mika Yoshimura, Mariko Kuse.

**Funding acquisition:** Hinako M. Takase, Itoshi Nikaido.

**Investigation:** Hinako M. Takase, Tappei Mishina, Tetsutaro Hayashi, Mika Yoshimura, Mariko Kuse.

**Methodology:** Tetsutaro Hayashi.

**Resources:** Hinako M. Takase, Mariko Kuse.

**Software:** Mika Yoshimura.

**Supervision:** Itoshi Nikaido, Tomoya S. Kitajima.

**Validation:** Tappei Mishina.

**Writing – original draft:** Hinako M. Takase, Tappei Mishina.

**Writing – review & editing:** Tomoya S. Kitajima.

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
