## [Decision Letter · Decision Letter 0]

9 Jun 2024

PONE-D-24-08138Transcriptomic signatures of WNT-driven pathways and granulosa cell-oocyte interactions during primordial follicle activationPLOS ONE

Dear Dr. Takase,

Thank you for submitting your manuscript to PLOS ONE. After careful consideration, we feel that it has merit but does not fully meet PLOS ONE’s publication criteria as it currently stands. Therefore, we invite you to submit a revised version of the manuscript that addresses the points raised during the review process.

We look forward to receiving your revised manuscript.

Kind regards,

Birendra Mishra, DVM, PhD

Academic Editor

PLOS ONE

Additional Editor Comments:

Please upload the images with higher resolution and ensure that the text in the figures is readable.

Reviewers' comments:

Reviewer's Responses to Questions

**Comments to the Author**

1. Is the manuscript technically sound, and do the data support the conclusions?

Reviewer #1: Yes

Reviewer #2: Yes

Reviewer #3: Yes

Reviewer #4: Partly

2. Has the statistical analysis been performed appropriately and rigorously? 

Reviewer #1: Yes

Reviewer #2: Yes

Reviewer #3: Yes

Reviewer #4: Yes

3. Have the authors made all data underlying the findings in their manuscript fully available?

Reviewer #1: Yes

Reviewer #2: Yes

Reviewer #3: Yes

Reviewer #4: Yes

4. Is the manuscript presented in an intelligible fashion and written in standard English?

Reviewer #1: Yes

Reviewer #2: Yes

Reviewer #3: Yes

Reviewer #4: Yes

5. Review Comments to the Author

Reviewer #1: The manuscript investigates the transcriptional changes of granulosa cells and oocytes during Primordial follicle activation (PFA) using Wntless (Wls) conditional knockout (cKO) mice, emphasizing the impact of the role of the WNT pathway in defining granulosa cell identity and their interactions with oocytes. This work contributes to our understanding of the mechanisms underlying the synchronous dormancy and activation of granulosa cells and oocytes during the PFA process. However, there are still some significant issues in the manuscript that require substantial revisions.

Major

This study utilizes the RamDA-seq, a full-length total RNA-sequencing method at the single-cell level, to examine GCs and oocytes. However, The choice of this method, along with its advantages over traditional single-cell RNA sequencing techniques, is not adequately justified or elaborated upon in the text. Does "full-length" sequencing imply the ability to analyze expression differences of isoforms between different cell clusters in control and Wls cKO mice?

The data analysis provided seems to lack depth. The absence of fundamental quality control measures and adjustments raises doubts about the reliability of the conclusions made.

The referencing order of the figures in the results section of the manuscript appears to be disorganized. Some figures are not well-associated with the corresponding text they are referenced to.

Minor

L38 The manuscript suggests, "The formation of primordial follicles occurs shortly after birth in mice," however, the process of primordial follicle formation actually begins shortly before birth, which also contradicts the statement in L96-97.

L54-57 FOXO3, GDF9, and other gene names should be italicized.

L79-85 The statements about what you have done following questions 2 and 3 might be better placed in a different context.

L166 In Figure 2, the colors representing the same cell types in both the control and Wls cKO groups should be consistent. This would allow for a clear visualization of the differences between the control and Wls cKO mice.

L177 Was the batch effect considered when integrating the single-cell data from Wls cKO and Ctrl groups.

L312-314 What is the reason for the notable difference in IRX3 expression between control and Wls cKO as shown in Figures 4E and 4F?

Reviewer #2: No concerns/conflicts of interest to address. No additional comments for the author, including concerns about dual publication, research ethics, or publication ethics. I agree with the above statements pertaining to questions 1-4.

Reviewer #3: In this manuscript, differences between ovarian granulosa cells and oocytes of wild-type and Wls cKO mice were analyzed at the single cell transcription level. The significant role of an autocrine function of WNT signaling in pre-GCs during primordial follicles activation, and the gene characteristics related to oocyte dormancy and activation were identified. However, the authors does not give clear conclusions on the three problems originally mentioned in the paper. This description should be carefully reviewed. Also, there are specific comments need to be addressed.

Specific Comments:

Major Issues:

1. In Fig. 2A, provide an explanation for why are there two different clusters of GC and oocyte in Wls cKO mice? How are altered granulosa cells and oocytes defined?

2. In Fig. 3A, there is almost no overlap in the distribution of UMAP between the GCs of wild type and Wls cKO mice, and it looks like the data were used ‘merge’ function rather than the ‘integrate’ function. Please include the distribution result of UMAP and PCA after direct merge and the PCA result after integrate.

3. In Fig .3D and 5F, authors should verified the expression of key DEG such as PDLIM4 and RPL27A by immunostaining in Wls cKO and control ovary at PD7 mice.

4. In Fig .6D, uthors should verified the expression and statistical results of NR1H2 and ENO1 by RT PCR in wild type and Wls cKO ovary.

Reviewer #4: Elucidation of the role and molecular mechanism of WNT-driven pathways in GC maturation, primordial follicle activation and folliculogenesis is important for understanding ovarian physiology and female reproductive biology. Whilst interesting, there are several concerns that need to be addressed or clarified.

1. Line 541-575, in the"single cells sorting", how authors to ensure only GCs were separated from the follicles? and what method was utilized to further identify the cells that were used for transcritomic analysis?

2. Line 166-172, Line 577-586, in the “Quality control” of scRNA-seq, principal component analysis (PCA) related results that can demonstrated the most important correlation or repeatability between the “single cells” samples should be supplemented.

3. To facilitate readers to understand the critical role of the WNT-signaling pathway and its molecular signature in regulating normal follicular development, the newly revealed key genes or signaling pathways involved in crosstalk with GCs, GC maturation and differentiation, and oocyte growth should be clearly indicated or provided, and one or two of them are suggested to be further verified.

6. PLOS authors have the option to publish the peer review history of their article (what does this mean?). If published, this will include your full peer review and any attached files.

Reviewer #1: **Yes: **Qiang Sun

Reviewer #2: No

Reviewer #3: No

Reviewer #4: No

---

## [Author Response · Author response to Decision Letter 0]

21 Sep 2024

Summary of responses to reviewers and editor:

We sincerely appreciate the reviewers' valuable feedback and suggestions, which have greatly helped improve the quality and clarity of our manuscript. We have carefully addressed all the points raised and have made the following significant revisions.

● A detailed overview of the quality control procedures has been provided.

● Analyses were conducted to confirm that batch effects were minimal.

● qPCR was performed to validate key gene expressions, confirming the scRNA-seq results.

● Immunofluorescent staining was repeated using PD7 samples, and the results have been updated accordingly.

The corresponding sections in the main text have been revised to reflect these updates. For a detailed, point-by-point response to all reviewer comments, please refer to the uploaded "Response_to_Reviewers" Word file.

---

## [Editor Report · Decision Letter 1]

29 Sep 2024

Transcriptomic signatures of WNT-driven pathways and granulosa cell-oocyte interactions during primordial follicle activation

PONE-D-24-08138R1

Dear Dr. Takase,

We’re pleased to inform you that your manuscript has been judged scientifically suitable for publication and will be formally accepted for publication once it meets all outstanding technical requirements.

Kind regards,

Birendra Mishra, DVM, PhD

Academic Editor

PLOS ONE

Additional Editor Comments (optional):

Thanks for responding the reviewers and editor's comments.
---

## [Editor Report · Acceptance letter]

10 Oct 2024

PONE-D-24-08138R1 

PLOS ONE

Dear Dr. Takase, 

I'm pleased to inform you that your manuscript has been deemed suitable for publication in PLOS ONE. Congratulations! Your manuscript is now being handed over to our production team.

Kind regards, 

on behalf of

Dr. Birendra Mishra 

Academic Editor

PLOS ONE